# Myasthenia Gravis: Pathogenic Effects of Autoantibodies on Neuromuscular Architecture

**DOI:** 10.3390/cells8070671

**Published:** 2019-07-02

**Authors:** Inga Koneczny, Ruth Herbst

**Affiliations:** 1Institute of Neurology, Medical University of Vienna, 1090 Vienna, Austria; 2Center for Pathophysiology, Infectiology and Immunology, Medical University of Vienna, 1090 Vienna, Austria

**Keywords:** myasthenia gravis, neuromuscular junction, AChR, MuSK, Lrp4, Agrin, autoimmunity, autoantibodies, history, immunopathogenesis

## Abstract

Myasthenia gravis (MG) is an autoimmune disease of the neuromuscular junction (NMJ). Autoantibodies target key molecules at the NMJ, such as the nicotinic acetylcholine receptor (AChR), muscle-specific kinase (MuSK), and low-density lipoprotein receptor-related protein 4 (Lrp4), that lead by a range of different pathogenic mechanisms to altered tissue architecture and reduced densities or functionality of AChRs, reduced neuromuscular transmission, and therefore a severe fatigable skeletal muscle weakness. In this review, we give an overview of the history and clinical aspects of MG, with a focus on the structure and function of myasthenic autoantigens at the NMJ and how they are affected by the autoantibodies’ pathogenic mechanisms. Furthermore, we give a short overview of the cells that are implicated in the production of the autoantibodies and briefly discuss diagnostic challenges and treatment strategies.

## 1. Clinical Aspects of Myasthenia Gravis

Myasthenia gravis (MG; from Greek: myos = muscle, asthenos = weakness, and Latin: gravis = severe) is an autoimmune disease of the neuromuscular junction (NMJ), and is considered a classic example of an antibody-mediated autoimmune disease. MG is a rare disorder, with an estimated prevalence of 70–163 per million for acetylcholine receptor (AChR) MG, and around 1.9–2.9 per million for muscle specific kinase (MuSK) MG [1,2,3]. Women are more often affected than men, with a female to male ratio of 3:1 for AChR MG and a ratio of 9:1 for MuSK MG [4]. The characterizing symptom is fatigable skeletal muscle weakness. Initial weakness often affects only ocular muscles, manifesting as ptosis (hanging of the eyelid) or diplopia (double vision). Most patients progress to generalized weakness, e.g., of limb muscles, within the first two years after disease onset. Other muscles that can be involved are bulbar muscles, which are necessary for speaking (leading to dysarthria), chewing and swallowing (causing dysphagia). Respiratory muscles can also be affected in up to 20% of cases with AChR MG, leading to a myasthenic crisis where patients need to be ventilated artificially. AChR MG can be further divided into several subgroups (see also Table 1): (1) Early-onset MG (EOMG) defines patients with an age of onset below 50 years, and are predominantly females with an onset in the 2nd and 3rd decade, frequently present with thymic hyperplasia; (2) late-onset MG (LOMG) with a higher fraction of male patients, often with an additional presence of striational antibodies; (3) thymoma-associated MG (TAMG), which affects approximately 10% of AChR MG patients; (4) ocular MG (OMG) with predominantly ocular symptoms; and (5) fetal or neonatal forms in which maternal autoantibodies pass the placenta. The passive transfer of antibodies against the adult AChR towards the fetus leads to a mild form of transient MG that passes weeks after birth. The symptoms include hypotonia, impaired sucking, swallowing, and breathing. Patients go into remission after days to months [5,6,7]. Antibodies against the fetal form of the AChR cause severe developmental defects and are a cause of arthrogryposis multiplex congenita [8,9,10,11].

A diagnosis of MG is primarily based on the observation of fluctuating skeletal muscle weakness. Initially, the patients often present with ocular muscle weakness with ptosis and diplopia that is not associated with sensory deficits. The diagnosis is then verified by (1) testing the serum for the presence of known myasthenic autoantibodies (discussed in Section 7 in more detail), (2) electrophysiological tests including single-fiber electromyography (SFEMG), and repetitive nerve stimulation (RNS), and (3) inhibition of acetylcholinesterase (AChE) e.g., with pyridostigmine or ice pack test, which usually improves the symptoms [12,13,14,15]. Although this is not a clinical review (excellent reviews focusing on clinical aspects and diagnosis can be found here: [16,17,18,19]), we would like to briefly discuss the electrophysiology of the NMJ in MG and its relevance for disease diagnosis.

ACh is released from synaptic vesicles upon the influx of Ca^2+^ into the nerve terminal. It is estimated that each vesicle contains 5000–10,000 ACh molecules and this amount is referred to as the “quantum” of neurotransmitter, which was first described by Fatt, Katz, and Miledi in 1952 [20,21]. A presynaptic action potential (AP) induces the release of about 20–200 quanta (in humans around 20 quanta compared to 200 in frogs) [22], which give rise to a depolarization of the muscle membrane, known as end-plate potential (EPP). The number of quanta released at a given NMJ during neuromuscular transmission is referred to as the “quantal content” of the EPP. The release of a single quantum results in small depolarizations (0.5–1 mV), the miniature EPPs (mEPPs). During a nerve impulse, mEPPs sum up to generate an EPP, which triggers an AP in the muscle fiber when the peak of the EPP reaches the critical threshold. In normal muscle, this happens with every nerve impulse. This reliable neurotransmission at the NMJ is the result of the ACh amount released being greater than that required to excite a muscle fiber. The release of an excess of neurotransmitter makes sure that neuromuscular transmission occurs under various physiological conditions. The term “safety factor” has been used to describe this excess [23]. Different definitions of the safety factor have been proposed and used (for more details see Wood & Slater [23]) but in general it defines the ratio of actual nerve-induced transmitter release to the amount of transmitter required to trigger an AP. Factors that affect the safety factor are the amount of released ACh, as well as the density of AChRs and Na^+^ channels at the postsynaptic membrane and the secondary synaptic folds [24]. The safety factor differs between species and muscles. Human intercostal muscle was reported to have a safety factor of 2 [25].

Neuromuscular disorders such as MG negatively affect neuromuscular transmission. Amplitudes of mEPPs and EPPs are greatly reduced, resulting in sub-threshold EPPs that fail to elicit muscle fiber excitation [25]. Recordings of compound motor action potentials (CMAPs) measures the sum of all APs triggered within particular muscle fibers and a failure of muscle fiber excitation can be detected by a decline of the amplitude. In addition to reduced EPPs, a decreased level of voltage gated sodium channels leads to an increased threshold to induce an AP in MG [26]. Upon RNS, this in turn leads to a reduced response to motor neuron stimulus by muscle APs, which can be measured as CMAP decrement [27]. In contrast, normal muscle shows no change in CMAP amplitude with RNS. In AChR-MG, the mEPPs are reduced, but the EPPs are still higher than expected, due to an increased quantal content [28]. This suggests a compensatory mechanism to overcome the reduced postsynaptic depolarization [29,30]. This has, however, not been observed in MuSK MG patients and MuSK MG animal models [31,32,33,34,35], as both the mEPP and EPP amplitudes were similarly reduced. Furthermore, MuSK antibodies also elicit presynaptic effects by blocking a retrograde signaling to the motor neuron, perhaps via disturbed anchoring of Lrp4.

MG (at least the subtypes with autoantibodies against AChR and MuSK) fulfils the Witebsky postulates that determine whether a disease is of autoimmune origin [36,37]. These include: 1) The presence of pathogenic autoantibodies or self-reactive T cells, 2) the reproduction of disease by the passive transfer of patient antibodies or T-cells to experimental animals, and 3) circumstantial evidence from clinical cues. Pathogenic mechanisms of serum or purified IgG from MG patients with AChR or MuSK autoantibodies have been identified, the passive transfer of patient serum or purified IgG has reproduced myasthenic weakness in experimental animals, and clinical cues for the autoimmune nature of the disease is the improvement of the symptoms after immunosuppression and after the removal of antibodies by plasmapheresis or B-cell depletion (discussed below in detail). MG with autoantibodies to Lrp4, Agrin, or ColQ do not yet fulfill the Witebsky postulates, since the pathogenic mechanisms of the antibodies are not entirely clear yet and, importantly, passive transfer animal models are still lacking.

MG is known as a classic example of a type II hypersensitivity reaction (following the Gell and Coombs classification). This means that IgG class autoantibodies target antigens expressed at the cell-surface or in the extracellular matrix and cause organ-specific damage. Most interestingly, MG can be caused by autoantibodies of different IgG subclasses. IgG1 and IgG3 subclass antibodies target the AChR, IgG1 and IgG2 target Lrp4, and interestingly, autoantibodies of the atypical IgG4 subclass target MuSK [38]. AChR and MuSK MG are considered archetypes for their respective kind of antibody-mediated autoimmune diseases. Pathogenic effects directly correlate to changes in NMJ structure and function, which greatly broadened our understanding of the physiology of the healthy NMJ, as well as the functioning of the immune system.

Patients with AChR MG frequently have thymic abnormalities, such as thymic hyperplasia (this is frequently associated with an early onset of disease) or thymoma. It is thought that the thymus plays a role in pathogenesis (discussed in Section 5.4). These patients respond well to the surgical removal of the thymus gland [39,40]. Patients with MuSK MG do not show thymic abnormalities and there does not seem to be a clinical benefit of thymectomy [41,42,43]. MuSK MG patients are also more severely affected by more frequent myasthenic crises (up to 40% of cases) [44,45,46,47]. Lrp4 MG patients are predominantly female (female to male ratio of 2.5:1) and have usually a generalized mild muscular weakness, unless additional antibodies to MuSK or AChR are present [48]. In an exceptional case, Lrp4 MG with severe muscle weakness manifested during infection with a new sequence isotype of *Leptospira interrogans* [49]. The role of the thymus in Lrp4 MG is unclear; one epidemiological study suggests the occasional presence of thymic hyperplasia [48]. A pilot study with four Lrp4 MG thymi showed no thymic abnormalities, but the positive clinical response in two of the patients still suggests a potential benefit of thymectomy [50].

## 2. History of MG

In 1672, myasthenia gravis was described for the first time by Thomas Willis, a physician from Oxford (Box 1) [51]. Further clinical studies in the 19th century by Samuel Wilks, Wilhelm Erb, and Samuel Goldflam led to the description of a disease characterized by fatigable skeletal muscle weakness. The very detailed reports contain a range of historically interesting observations. In one paragraph, a patient was described to have fallen in a shallow pond while cleaning the dishes, and needing rescue due to weakness. Treatment suggestions included taking tonics, specifically those of arsenic, chinin, iodine, iron, and digitalis, as well as warm baths, cold showers, remedial exercises, and daily “galvanisation” of the spinal cord with a current of 2–3 milliampere. The reports also contained very astute clinical observations, such as the fluctuating nature of the weakness and the patterns of muscle weakness [52,53,54]. The disease was thus named Erb–Goldflam disease, until 1895 when Friedrich Jolly coined the term “myasthenia gravis pseudoparalytica” [55], and showed by repetitive nerve stimulation that the innervated muscle had a decrease in response, thus proving that neuromuscular transmission was affecting these patients. Ground-breaking work by Otto Loewi [56] demonstrated the chemical nature of the synapse and in 1934, Henry Dale discovered that acetylcholine (ACh) release is essential for neuromuscular transmission [57,58]. In 1934, Mary Walker discovered that an injection of physostigmine and neostigmine, back then already known to prevent the break-down of ACh, improved facial weakness in a MG patient. This led to the conclusion that MG is a “curare-like poisoning of the motor end-organs” [59].

Box 1**Original quote by Thomas Willis (*translated by the English poet Samuel Pordage in 1688*)**.“…those labouring with a want of Spirits, who will exercise local motions, as well as they can, in the morning are able to walk firmly, to fling their Arms hither and thither, or to take up any heavy thing; before noon the stock of the Spirits being spent, which had flowed into the Muscles, they are scarce able to move Hand or Foot. At this time I have under my charge an Honest Woman, who for many years hath been obnoxious to this sort of spurious Palsie, not only in her Members, but also in her tongue; she for some time can speak freely and readily enough, but after she has spoke long, or hastily, or eagerly, she is not able to speak a word, but she becomes mute as a Fish, nor can she recover the use of her voice under an hour or two.”

The characteristic histological malformations of the NMJ were first described by Coers and Desmedt in 1959 [60]. The idea that autoantibodies could damage tissues and cells belonging to their own body was first formulated at the end of the 19th century, when Paul Ehrlich first hypothesized that so-called “anti-toxins” could be directed against self-structures in the body (reviewed by [61]). In 1960, Simpson and Nastuk hypothesized that MG might be an autoimmune disease [62,63]. This theory was proven true in 1973, as Patrick and Lindstrom discovered (by accident no less) the pathogenic effect of AChR antibodies, when they immunized rabbits with the aim to produce AChR antibodies. The resulting antibodies cross-reacted with the rabbit’s own AChRs, and the animals developed a MG-like muscle weakness. Soon afterwards, AChR antibodies were found in MG patients using a radioimmunoprecipitation assay [64,65]. The passive transfer of patient antibodies to mice also reproduced the disease, proving the pathogenicity of AChR autoantibodies [66].

These studies were made possible by the isolation of α-bungarotoxin, which is a snake toxin that binds AChR irreversibly [67,68] and the isolation of AChRs from the electric organs of Torpedo fish [69]. AChR antibodies were found to block the function of AChRs [70] and Fambrough and Drachman discovered a markedly reduced number of AChRs in the muscle biopsies of MG patients [71]. Furthermore, IgG and complement depositions were found at the NMJ of these patients [72,73]. AChR MG and its three key pathogenic mechanisms are well studied today and are discussed in more detail in Section 5.1. New extracellular autoantibody targets were only discovered in the new millennium, and these are to date MuSK [74], Lrp4 [75,76,77], Agrin [78,79], and ColQ [80]. It seems that the discovery of new antibody targets in MG is traditionally met with disbelief and doubt. Initial studies in MuSK MG showed an absence of complement deposition and a loss of AChR in patient biopsies (taken from intercostal muscles), which led to doubt that these antibodies are pathogenic. However, the pathogenicity of MuSK autoantibodies has been demonstrated in experimental animals by active immunization with MuSK antigen and by passive transfer of patient serum and IgG [33,81,82,83], including the passive transfer of purified IgG4 from patient serum [84]. Similarly, Lrp4 and Agrin antibodies as new pathogenic entities in MG are met with skepticism, since key experiments to prove their pathogenicity are still lacking (as discussed in Section 5.3). To date, there are active immunization models for Lrp4 MG [85,86,87] and Agrin MG [88] available, but unfortunately a passive transfer of patient serum or IgG to experimental animals, which would prove the pathogenicity of these autoantibodies, is still lacking. The pathogenic mechanisms have been investigated, but remain unclear. There are no studies on the pathogenicity of ColQ antibodies, neither in vitro nor *in vivo*.

Autoantibodies against intracellular antigens were also discovered in MG patients, among these are autoantibodies to three striational antigens, ryanodine receptor, titin and Kv1.4, and cortactin [89,90,91,92]. These antibodies are considered as non-pathogenic but have diagnostic value as biomarkers [38,93,94,95,96], with striational antibodies being associated with the presence of a thymoma [97,98]. Despite these advances, a population of seronegative MG patients with no known autoantibodies remains, and further studies in antigen discovery are required to identify the pathogenic autoantibodies in this population. An excellent review on the history of MG for further reading can be found here [99].

## 3. Antigens and Their Structure

### 3.1. AChRs

More than 60 years ago, Del Castillo and Katz discovered that AChRs were localized to the NMJ [100]. Ultrastructural investigations of membrane preparations of the *Torpedo californica* electric organ led to a description of the AChR as a hexagonal structure, proposed to be a ring composed of six receptor subunits [101], which later turned out to be a pentamer composed of five receptor subunits [102]. AChRs were purified and biochemically characterized by a great many of studies, again mostly in the Torpedo electric organ (extensively reviewed in [103,104]). AChR genes were cloned subsequently in the 1980s, again first from Torpedo and later from rodents and humans [105,106].

AChRs are member of a superfamily of neurotransmitter-gated ion channels, each comprised of five homologous subunits arranged around a central ion channel. Seventeen AChR subunits have been cloned and are subdivided into four classes. Class I–III represent neuronal AChR subunits and class IV include muscle AChRs.

AChR subunits show 35–50% sequence homology in the N-terminal region, are glycosylated, and share structural features (Figure 1A) [107]. Three highly conserved and mainly α-helical transmembrane domains (M1–M3) encompass between the large extracellular domain and the cytoplasmic domain (containing one α-helix). A fourth α-helical transmembrane domain (M4) crosses back to the extracellular space creating a short (10–20 amino acids) extracellular sequence. The N-terminal extracellular portion is organized around a β-sandwich core and the cytoplasmic domains of AChR β and δ contain a regulated phosphotyrosine site, which is important for cytoskeletal anchorage [108,109]. Muscle AChRs have the composition α_2_βδγ in embryonic muscle or α_2_βδε in adult muscle [103]. AChR subunits are organized like barrel staves around the central ion channel in the order αγαδβ (Figure 1B). ACh binding sites are present at the subunit interfaces: α_γ_-γ or ε and α_δ_-δ. The pore-lining helices of the closed channel create a hydrophobic narrowing in the membrane, which acts as the gate. The transition to an open, active state is facilitated by the occupation of both agonist binding sites. This triggers a reorganization of the ligand-binding domain that displaces the extracellular region of the β subunit. The β-subunit helices tilt outward and destabilize the arrangement of pore-lining helices, expanding the pore asymmetrically and enhancing its polarity in the gate region [110,111,112]. This paired allosteric transition changes the structure from a tense (closed) state toward a more relaxed (open) state.

The N-terminal region of AChR α represents the main immunogenic region (MIR) [113,114,115]. The MIR is a cluster of overlapping epitopes rather than one single epitope and epitopes are conformation dependent. Half of all MG patients generate autoantibodies against the MIR. The MIR is angled outward from the central axis of the AChR, which prevents the cross-linking of two α subunits within an AChR, and instead induces the cross-linking of adjacent AChRs (Figure 1C). MIR-specific antibodies do not interfere with the binding of ACh or α-bungarotoxin to the ACh binding site, nor do they allosterically affect the AChR function.

### 3.2. Agrin

In the 1970s, Jack McMahan proposed that regulatory signals for forming and maintaining synaptic differentiation may be present within the synaptic extracellular matrix, or basal lamina of the NMJ. This hypothesis was confirmed by a set of regeneration experiments using the musculus pectoralis of the frog. McMahan and colleagues demonstrated that pre- as well as postsynaptic differentiation is controlled by factors localized in the synaptic basal lamina [116,117,118]. The electric organ of *Torpedo californica* was used to purify an activity, which stimulated the clustering of AChRs in cultured muscle cells [119]. This led to the identification of a protein named Agrin (from Greek “ageirein”, to gather together) [120].

The Agrin gene encodes for a ~200 kDa protein with multiple protein binding domains (Figure 2A). The protein is expressed in many tissues and cell types, exists in different splice variants, and is post-translationally modified by glycosylation [121]. The Agrin isoform, which is important for NMJ development, is made by motor neurons, hence being called neuronal Agrin, and contains a N-terminal laminin binding domain required for its anchorage to the basal lamina and an 8, 11, or 19 amino acid insert in the B/Z splice site within the C-terminal portion of the protein, which is essential for AChR clustering [122]. The carboxy-terminal half of the protein with three laminin-G-like (LG) domains and four epidermal growth factor- (EGF) like domains is fully active in AChR clustering when added to cultured muscle cells [123]. The crucial B/Z splice site is located in the N-terminus of the most C-terminal LG domain (LG3). LG3 adopts a 14-strand β jellyroll fold that is homologous to the LNS (laminin, neurexin, and sex hormone-binding globulin-like) domain [124,125]. An 8 amino-acid B/Z region was found to form a long loop, which represents the primary interface between Agrin and Lrp4. A lack of the B/Z loop or mutation of critical residues at the tip of the loop interferes with Lrp4 binding [125]. Although Agrin LG3 is monomeric in solution, it dimerizes in a tetrameric Agrin–Lrp4 complex. The Agrin–Agrin dimer interface lies close to a Ca^2+^-binding site in Agrin, which may enforce the local conformation of Agrin and foster dimer formation, thereby enhancing the Agrin–Lrp4 interaction.

### 3.3. Lrp4

It took more than 20 years to identify Lrp4 as the Agrin receptor at the NMJ. After the discovery of MuSK, researchers very quickly realized that Agrin does not directly bind to MuSK [126]. The hypothetical molecule MASC (myotube-associated specificity component) was introduced being either a muscle-specific modification, co-receptor, or co-ligand. A recessive screen for genes involved in early mouse development identified two mutant alleles of Lrp4, a member of the low-density lipoprotein receptor family [127]. These *Lrp4^−/−^* mice had a variety of developmental malformations including digit and craniofacial defects, kidney hypoplasia, and agenesis, but in addition, the *Lrp4* mutants died at birth due to respiratory failure.

Lrp4 is expressed in multiple tissues in the mouse, and is therefore important for the proper development and morphogenesis of different organs such as limbs, lungs, and kidneys [127]. Lrp4 is a single-pass transmembrane protein with a large extracellular domain that contains eight low-density lipoprotein receptor domain class A (LDLa) repeats, two EGF-like domains, and four β-propeller (BP) domains, each of which is fused together with an EGF-like domain (Figure 2B). Neuronal Agrin binds predominantly to the first BP domain, although the last few LDLa repeats contribute to Agrin-binding [128]. The crystal structure of the Agrin-Lrp4 complex showed that the central part of BP1 represents a six-bladed β-propeller domain (conserved in the LDLR family) with two slightly concave surfaces perpendicular to the β-propeller’s central axis [125]. One surface is enclosed by the two flanking EGF-like domains, the second surface is involved in Agrin binding via the B/Z loop (as described above). Crystal structures of close relatives of Lrp4 such as Lrp6 demonstrated the use of the same surface for ligand binding. The Agrin–Lrp4 dimers further assemble into a tetrameric complex via interfaces between Agrin and Lrp4 (no Lrp4 dimers were observed), which is required for MuSK activation. The third BP domain as well as the fourth and fifth LDLa repeats are critical for binding of Lrp4 to MuSK (via the first Ig-like domain, described below) [128]. The transmembrane and intracellular domains are not required for MuSK activation by Agrin.

### 3.4. MuSK

MuSK was first identified as a novel Trk-related receptor tyrosine kinase (RTK) enriched in the electric organ of *Torpedo californica* [129]. Later, it was isolated as a gene product that was highly induced during muscle denervation in mice [130]. Its expression was initially thought to be restricted to skeletal muscle, hence the name muscle-specific kinase, but MuSK was subsequently also found in other tissues and cells [131,132,133,134].

MuSK presents a single member subfamily within the family of RTKs. MuSK contains three Ig-like (Ig) domains and one cysteine-rich domain (CRD), also known as Frizzled (FZ)-like domain in the extracellular region. The transmembrane region is followed by a short juxtamembrane region, a tyrosine kinase domain, and an eight amino-acid C-terminal sequence. With the exception of mammals, all other vertebrate species, including zebrafish and chicken, carry a Kringle domain between the CRD and the single transmembrane helix. The Ig1 domain is crucial for MuSK homodimerization as well as Lrp4 binding [128,135]. In particular, a solvent-exposed region in Ig1 of MuSK (around Ile 96) is required for MuSK to bind Lrp4 (Figure 2C). The function of the CRD is unresolved, as it is not required for Agrin-induced MuSK activation, but was shown to mediate Wnt-induced MuSK activation and AChR clustering in cultured muscle cells as well as muscle pre-patterning in zebrafish [136,137,138]. The MuSK CRD binds Wnt proteins *in vitro*, but MuSK activation and AChR clustering by Wnt requires Lrp4 expression in muscle cells [137,139]. Similarly, genetic studies using mice expressing MuSK ΔCRD produced conflicting results [140,141]. Likewise, the role of the Kringle domain in non-mammalian MuSK is unclear. The kinase domain is composed of a N-terminal lobe with a five-stranded β sheet and one α helix, and a predominantly α helical C-terminal lobe [142]. The N-terminal lobe contains residues important for ATP binding and the C-terminal lobe includes the activation loop and the catalytic loop. Mutation of tyrosines in the activation loop as well as the mutation of Lys608 in the C-terminal lobe interfere with MuSK activation [136,143,144]. Tyr553 is located in the juxtamembrane region and represents a major phosphorylation site within a NPYX motif, which, upon phosphorylation of Tyr553, recruits Dok-7 [142,145,146]. The mutation of Tyr553 and failure of autophosphorylation inhibits MuSK activation and blocks Dok-7 binding [144]. The C-terminal sequence is disordered in the crystal structure and is therefore accessible for protein binding, but functional studies have shown that this region is dispensable for the MuSK function [136].

### 3.5. ColQ

ColQ is localized to the extracellular matrix and is crucial for the anchoring of AChE to the NMJ [147]. It is a collagen in which the central collagenous domain is flanked by specific N- and C-terminal peptides. Trimers of ColQ conjoin in a triple helix tail. There are three forms of AChE generated by different splicing and posttranslational modifications: AChE_T,_ which is the main isoform found in the muscle, brain, and other tissues that is a relevant splice variant at the NMJ, AChE_H,_ which is expressed in erythrocytes, and AChE_R,_ which is not expressed in humans. Tetramers of catalytic AChE_T_ subunits associate with the collagen tails according to levels of AChE present [148]. At the NMJ, most of the enzyme is present as A_12_ forms, which have all three collagen tails filled with tetramers. *ColQ*-deficient mutant mice lack clusters of AChE at the NMJ and congenital myasthenic syndromes associated with AChE deficiency are caused by recessive mutations in ColQ [149,150]. More recently it was demonstrated that MuSK interacts with ColQ, thereby conferring synaptic localization of AChE [151].

## 4. Antigen Function during NMJ Development

### 4.1. The Agrin/Lrp4/MuSK Signaling Complex

Agrin, Lrp4 and MuSK represent a functional unit that constitutes the primary scaffold of the developing NMJ that initiates post- as well as presynaptic differentiation. MuSK is activated by the neuron-specific isoform of Agrin [126], a heparansulfate proteoglycan made by motor neurons and deposited in the basal lamina of the synaptic cleft [152]. Agrin does not bind MuSK directly, but interacts with Lrp4 [153,154]. Lrp4 and MuSK are pre-assembled in the absence of Agrin but the activation of MuSK is only induced in the presence of Agrin, which forms a tetrameric complex with Lrp4 [125]. Agrin binding to Lrp4 increases MuSK-Lrp4 interaction by a yet unknown mechanism. Upon binding, the Lrp4/MuSK tetramer presumably rearranges in a way that leads to dimerization and subsequent autophosphorylation of MuSK. Recruitment of Dok-7 further enhances MuSK dimerization resulting in the full activation of the MuSK kinase [145,155,156]. The downstream signaling cascade induces the formation of the NMJ, including postsynaptic differentiation characterized by the accumulation of AChRs at synaptic sites and presynaptic differentiation illustrated by the development of active zones [157] (Figure 3A). In accordance with this model, mice lacking *Agrin*, *MuSK*, or *Lrp4* fail to form NMJs and consequently die at birth due to respiratory failure [127,158,159]. The AChR expression is normal, but very few nerve contacts (in *Agrin^-/-^* mice) or no contacts (in *MuSK^-/-^* and *Lrp4^-/-^* mice) are associated with AChRs or other postsynaptic specializations. In addition, motor axons keep on growing and do not form arborized nerve terminals. Synaptic proteins like AChE, rapsyn, neuregulin-1, and its receptors, the ErbBs, which are usually clustered at NMJs, are uniformly distributed in muscle from mice lacking Agrin, MuSK, or Lrp4 [127,158,159]. Similarly, synapse-specific transcription in nuclei underlying NMJs is absent in these mice since genes, such as the *AChR* subunits genes, which are transcribed equally in synaptic and non-synaptic nuclei.

### 4.2. Prepatterning and NMJ Formation

Aneural AChR clusters were first described 20 years ago [160,161,162]. This so-called pre-patterning of the muscle depends on MuSK and Lrp4 and happens prior to the arrival of motor neurons. In contrast, *Agrin* mutant mice develop aneural AChR clusters, but these are dispersed upon innervation [160]. It was therefore postulated that Agrin-independent MuSK activation induces pre-patterning and that nerve-derived Agrin acts as a stabilizer of pre-existing AChR clusters. Agrin is anchored to the basal lamina through the N-terminal laminin binding domain and binds to Lrp4, which induces the dimerization and autoactivation of MuSK. The MuSK signaling then induces pre- as well as postsynaptic differentiation. Upon innervation, MuSK expression is rapidly downregulated and MuSK protein is specifically localized to the postsynaptic muscle membrane [130]. In addition, tyrosine phosphates such as Shp2 counterbalance MuSK activation [163]. These regulatory mechanisms and the spatially restricted activation by Agrin are thought to ensure localized signaling at synaptic sites.

### 4.3. NMJ Maturation and Maintenance

For precise and refined muscle contraction, NMJ maturation plays a critical role. The size of the NMJ increases and its morphology and molecular composition changes to warrant a rapid and efficient neurotransmission [157]. At the pre-synapse, multiple inputs are eliminated to one nerve terminal per postsynaptic site, and a number of synaptic vesicles and active zones increase. The muscle membrane invaginates to form post-junctional folds, proteins become compartmentalized along these folds, the AChR subunit composition changes from α_2_βδγ to α_2_βδε, and AChRs become greatly enriched at the crest of postsynaptic folds (more than 10,000 receptors/μm^2^). Morphology and high AChR density are maintained throughout most of an organism’s adult life. The molecular mechanisms that regulate maturation and maintenance are not well understood. However, lasting Agrin/Lrp4/MuSK signaling is also required for NMJ maturation and maintenance. The elimination of Agrin, MuSK, or Lrp4 in adult mice leads to NMJ disassembly [164,165,166,167]. Proteolytic cleavage of Agrin was shown to influence NMJ remodeling and maturation [168]. *In vitro*, it was shown that the ECM-induced topologically complex AChR cluster depends both on MuSK kinase activity and the integrity of the extracellular domain of MuSK [169]. Furthermore, mutations in *MUSK*, *AGRN*, or *LRP4* cause congenital myasthenic syndromes associated with NMJ fragmentation, loss of AChRs, and impaired neurotransmission and muscle function [170,171].

### 4.4. NMJ Aging

During aging, muscle fiber number, size, and types change and become infiltrated with adipocytes and connective tissue [172]. Associated with these alterations, NMJ fragmentation, reduced AChR density, and denervation have been observed [173]. In aged human subjects, motor neurons are lost, whereas in mice the presynaptic phenotype represents more variables (depending on the muscle type) [172,173,174,175]. Further controversial questions are whether changes in NMJ morphology are associated with a decline in neurotransmission and whether NMJ degeneration contributes to aging or results from it [176]. Given the changes in NMJ morphology, in particular fragmentation and the loss of AChRs, it is tempting to speculate that these changes might contribute to LOMG. The mechanisms that lead to NMJ decline in aging organisms are not well understood. As described above, NMJ fragmentation and disassembly was observed in adult mice lacking Agrin or Lrp4 [164,167]. Furthermore, enhanced proteolytic cleavage of Agrin destabilizes NMJs and induces denervation [177].

## 5. Pathogenic Effects and Origin of Autoantibodies

### 5.1. AChR Antibodies

Approximately 85% of patients have autoantibodies against the AChR. These antibodies mainly belong to the IgG1 and IgG3 subclass (see additional information in Box 2) [178,179,180], and many recognize the MIR [181,182,183,184]. The AChR antibodies induce pathogenicity by three main mechanisms (Figure 3B): (1) Activation of the classical complement cascade [185]. Here, C1q binds to the Fc region of the AChR antibodies, followed by the breakdown of C3 into C3a and C3b, which, as downstream products of the complement cascade, initiate the formation of the membrane attack complex (MAC). C3 and patient IgG, as well as MAC, can be found colocalized with AChR at the NMJ of patients [72,73,186,187]. The patient serum is depleted of early complement components C3 and C4 while terminal complement components are increased [188,189]. The complement attack leads to membrane lysis and severe damage of the postsynaptic apparatus, including simplified junctional folds, debris in the intrasynaptic space, and a loss of AChRs, as well as voltage-gated sodium channels from the synapse [186,190,191]. Furthermore, patient serum induced a complement-mediated lysis of rat myotubes *in vitro*. [192]. Complement mediated damage is a major cause of the pathogenicity of AChR antibodies, which can be demonstrated in animal models of MG. Soluble complement receptors improved symptoms in a passive transfer rat model [193], the inhibition of the complement cascade by suppressing the expression of C2 or C5 significantly improved clinical symptoms in a mouse (C2) [194] and rat (C5) [195] model of MG as did the linking of a complement regulator to the motor endplate via coupling to a recombinant AChR antibody [196]. Inhibitors of late complement component C5 are by now in clinical use (Section 6.1). (2) Binding and cross-linking of AChRs by antibodies (many of which target the MIR [197]) may lead to endocytosis and therefore a loss of AChR densities. The normal half-life of AChRs at the mature synapse is eight to 11 days [198]. Patient IgG cause an increased turnover and degradation of AChRs in skeletal muscle cells [199,200,201,202,203]. The mechanism requires the divalent binding of an antibody, as Fab fragments (derived from patient IgG) were unable to induce endocytosis, unless a secondary antibody was used to cross-link the Fabs [201]. (3) The direct inhibition of the function of the AChRs by preventing the binding of ACh or blocking the channel [70,204,205,206,207].

Box 2**Antibody structure and function**.

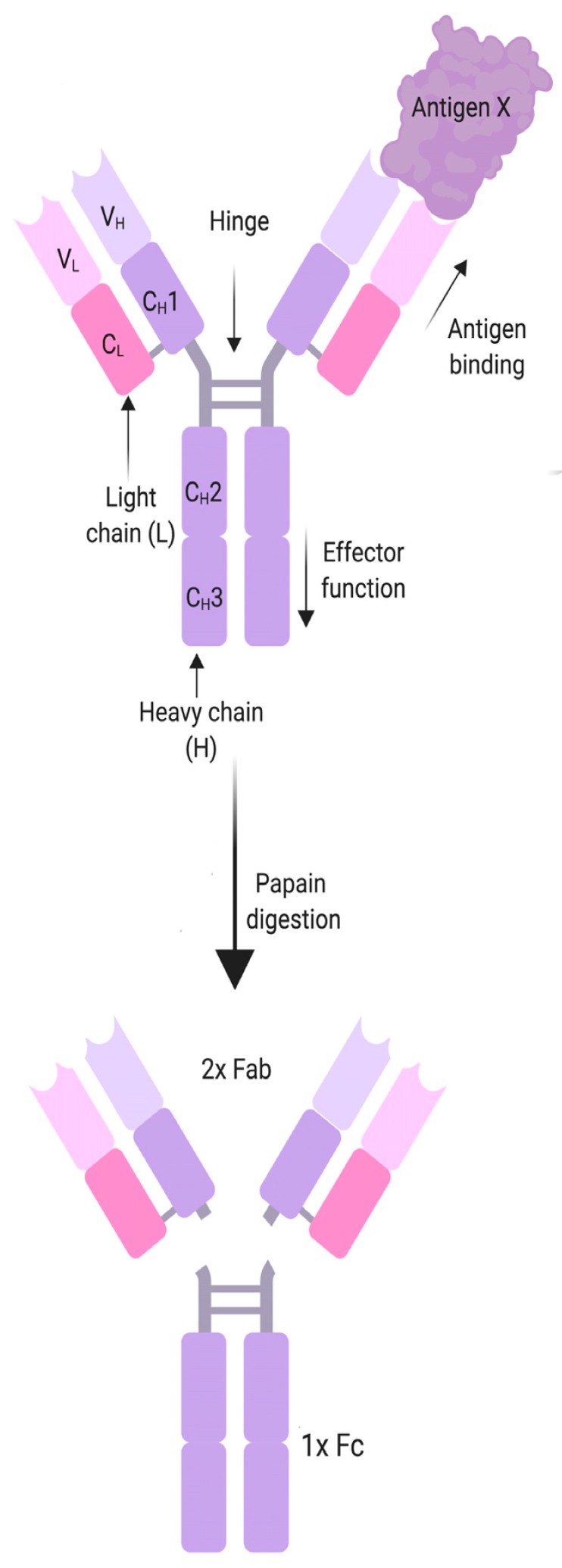

MG antibodies belong to the IgG class, which can also be divided into the Fc and the Fab region. Historically these names derive from the two breakdown products generated when IgG is digested with papain. The “Fab” part, for “Fragment, antigen binding”, is the part of the antibody that contains the variable regions of the heavy and light chains (V_L_, V_H_), which binds the antigen, as well as the first constant domain (C_H_1, C_L_). Fc stands for “fragment, crystallized”, and comprises most of the constant region of the two heavy chains of an antibody (C_H_2-C_H_3), and determines the antibody class and subclass, whether the antibody is membrane bound or soluble, and the effector mechanisms of the antibody. These may include the activation of the classical complement cascade, antibody-dependent cellular cytotoxicity, opsonization, blocking enzymes or receptors or formation of immune complexes. Many autoantibodies belong to the IgG1 and IgG3 subclasses that bind C1q and activate the classical complement pathway, and that also bind to activating Fcγ receptors on immune cells leading to their activation. IgG1, 2 and 3 also cross-link antigens, forming either immune complexes with soluble antigen, or causing endocytosis of transmembrane proteins. In recent years a range of autoantibodies associated with IgG4 subclass antibodies were discovered. Due to structural differences in the Fc region, IgG4 does not bind C1q or activating Fcγ receptors and is bi-specific, and therefore has an “anti-inflammatory” effect thought to counteract overshooting immune responses after chronic antigen exposure. IgG4 autoantibodies are usually pathogenic by a blocking mechanism that affects enzyme or receptor function or disrupts protein-protein interaction. Figure created with BioRender.

The loss of AChRs by (1) and (2) has consequences for neuromuscular transmission as reduced AChR densities lead to reduced sensitivity for acetylcholine, reduced mEPPs, and reduced EPPs, and an overall reduction of the safety factor (see Section 1). Interestingly, it was found that patients with AChR antibodies show an increased quantal content [28], which suggests a retrograde signal from the muscle to the nerve in an attempt to compensate for reduced AChR levels by an increase of ACh release, though it remains unclear by which retrograde signal this occurs [29,30].

While most AChR antibodies target the AChR α subunit, there are individuals with antibodies against the fetal γ subunit. The γ subunit is only expressed during the first 30 weeks of life, after which it is exchanged by the adult ε subunit with the exception of the extraocular muscle, where AChR γ subunit expression is maintained [208]. Therefore, adults with these antibodies do not develop MG. When a healthy, pregnant woman produces anti-γ subunit antibodies, they can be transferred through the placenta and to the embryo. Here, the antibodies cause a fetal AChR inactivation syndrome, which leads to the reduced movement of the fetus. This has dire developmental consequences, the new-born children present with arthrogryposis multiplex congenita, a developmental disorder hallmarked by multiple joint contractures and profound respiratory impairment that may lead to severe disabilities or fetal death [8,9,10,11,209].

### 5.2. Antibodies to MuSK

MuSK antibodies behave fundamentally different from AChR antibodies. They belong to the IgG4 subclass [210,211], which does not activate the classical complement system or immune cells, due to structural differences in the Fc region (see Box 2). Furthermore, IgG4 can undergo Fab-arm exchange, which is the exchange of IgG4 half-molecules and the generation of bi-specific antibodies (reviewed in detail here [212]). This has also been demonstrated for MuSK autoantibodies [213]. In fact, an estimate of 99% of IgG4 and MuSK IgG4 is therefore bi-specific, or functionally monovalent, which means they cannot cross-link an antigen of the same kind, excluding antigenic modulation as pathogenic mechanism. Epitope mapping studies [74,210,214,215,216] suggest that MuSK antibodies bind to the first two Ig-like domains, and in a subset of patients also to the CRD domain, but the first Ig-like domain is considered as the predominant target (Figure 3C). This is interesting because this domain covers the region around Ile96 [128], which is required for the interaction of MuSK with Lrp4. And indeed, it was found that MuSK antibodies of the IgG4 subclass exclusively led to a block of Lrp4-MuSK interaction [211,217]. This in turn caused a reduction of MuSK phosphorylation [217]. Moreover, purified IgG4 from MuSK MG patients, as well as monovalent Fab fragments derived from patient IgG, led to a reduction of newly formed Agrin-induced AChR clusters [211]. Taken together, this demonstrates that the MuSK antibodies interrupt the Agrin-Lrp4-MuSK-Dok-7 signaling axis, causing reduced densities of AChRs at the synapse and therefore defects in neuromuscular transmission.

Notably, IgG1–3 antibodies do not have the same effect on MuSK-Lrp4 interaction, but were still able to reduce Agrin-induced clustering of AChR in C2C12 myotubes [211], suggesting an additional pathogenic mechanism. One effect could be a block of MuSK dimerization, as the first Ig-like domain also contains a hydrophobic patch around Leu38, which is essential for MuSK dimerization and activation [135]. However, one study did not find reduced MuSK dimerization by patient antibodies [217], and divalent commercial antibodies against MuSK were found to rather induce MuSK dimerization and activation by cross-linking [218]. A recent study showed that, interestingly, the effect depends on antibody valency [219]. Recombinant divalent MuSK antibodies, derived from MuSK MG patient B-cells, led to the phosphorylation of MuSK and induced Agrin-independent AChR clustering, presumably by dimerization, while monovalent Fab fragments from the same antibody clone reduced MuSK phosphorylation and AChR clustering. This suggests that overall there may be (at least) two competing mechanisms at work: 1) MuSK IgG4 (monovalent) blocks Lrp4-MuSK interaction and MuSK activation, while 2) divalent MuSK IgG1–3 antibodies induce ectopic, Agrin-independent MuSK dimerization and activation thereby recruiting AChRs to extrasynaptic sites (Figure 3C).

Patient-derived MuSK antibodies, as well as Fab fragments, are also able to disperse pre-existing AChR clusters that were induced by overexpression of Dok-7 [211,220]. It is unclear how Dok-7-induced AChR clusters can be affected, since Agrin, and probably also Lrp4, are not involved in the formation of these AChR clusters. Possible mechanisms could be a disruption of MuSK-Dok-7 interaction, perhaps induced by conformational changes in MuSK, or a disruption of Dok-7-induced MuSK dimers, either by preventing MuSK-MuSK interaction itself, or by inducing new MuSK dimers by cross-linking, and in this process also disrupting pre-existing MuSK dimers. 

Interestingly, MuSK antibodies also affect the retrograde signaling to the motor neuron. In patients and animal models, presynaptic abnormalities were found, as well as a reduced quantal release of AChE [31,32,33,34,35]. It is unclear which retrograde signaling pathway could be modified by the antibodies, but it is suspected that the disruption of Lrp4-MuSK interaction may play a role, perhaps by a loss of anchoring of Lrp4 to the synapse, as Lrp4 contributes to a retrograde signal from the muscle to the motor neuron [221,222].

### 5.3. Antibodies to Lrp4, Agrin, and ColQ

Lrp4 itself is also a target for autoantibodies [48,75,76,77]. It is not clear yet whether Lrp4, Agrin, and ColQ antibodies are pathogenic, since relevant experiments are still missing. The pathogenic mechanisms of Lrp4 are under investigation, but one might speculate that Lrp4 autoantibodies could interfere either with binding to Agrin or with binding to MuSK. Indeed, a recent study showed a disruption of the Lrp4-Agrin interaction by patient serum using immunoprecipitation of tagged Agrin and co-immunoprecipitation and quantification of Lrp4 [76], and a second study showed a disruption of Agrin binding to immobilized Lrp4 on ELISA plates by Lrp4 MG patient serum [75]. In line with a non-functional Agrin-Lrp4-MuSK signaling axis, Lrp4 MG patient serum also reduced Agrin-induced AChR clustering in C2C12 mouse myotubes [75,77,85]. Reduced Agrin-induced AChR clustering and reduced Agrin-induced MuSK phosphorylation could be reproduced in active immunization mouse models [85,86].

Since the antibodies are of an IgG1/2 subclass, further mechanisms might be possible. They might activate the complement system, and Lrp4 antibodies from immunized mice or rabbits do activate complement [86,87], but this mechanism has not been demonstrated in MG patients yet. Active immunization of mice with extracellular Lrp4 led to the production of complement-fixing mouse IgG2 antibodies to Lrp4 (equivalent to human IgG1). The mice had IgG and complement C3 deposits at the NMJ, and cells derived from the lymph nodes produced pro-inflammatory cytokines in response to immunogen stimulation, similar to an active immunization model of AChR MG [87]. These data require validation by studies with human patient serum, since the active immunization of the mice may not reflect the physiological situation in the patient. The patient antibodies could theoretically also cross-link Lrp4 and cause endocytosis and degradation of autoantibodies, similar to AChR antibodies. Fragmented AChR clusters, distorted axon terminals, and an abnormal NMJ ultrastructure have been shown in an active immunization mouse model, but not yet in experiments with patient antibodies [86]. Active immunization models also showed postsynaptic abnormalities and reduced mEPPs, as well as a CMAP decrement similar to MG patients, myasthenic weakness, and interestingly, presynaptic changes [85,86,87]. This (taken together with presynaptic changes in MuSK MG) is further evidence that Agrin-Lrp4-MuSK interaction is relevant for retrograde signaling to the motor neuron, though it is not understood how. Interestingly, Lrp4 antibodies from an active immunization mouse model also affected Agrin-independent AChR clustering, which is an observation that was also made with patient derived MuSK antibodies [85,211] and suggests that there may be further mechanisms at work in the development and stabilization of AChR clusters, perhaps by yet unknown binding partners of MuSK and Lrp4 that act independent of Agrin.

There are only a few studies on Agrin autoantibodies, but in one study, patient serum was found to inhibit Agrin-induced MuSK phosphorylation in C2C12 myotubes, suggesting a blocking of Lrp4-Agrin interaction [78], and in line with these findings the active immunization of mice with neural (but not muscle) Agrin-induced myasthenic weakness and fragmented synapses [88]. These results suggest that the autoantibodies may be directed against the B/Z loop located in LG3, which would explain the observed lack of MuSK activation due to an impairment of Agrin-LRP4 interaction. The pathogenic mechanisms of antibodies against ColQ have not been studied yet. Importantly, to prove the pathogenicity of Lrp4, Agrin, and ColQ autoantibodies, it is necessary to passively transfer patient derived serum or IgG to experimental animals and to show that myasthenic weakness could be reproduced in the animal model.

### 5.4. Autoantibody Production in MG

The identification of the autoantibodies’ source is essential as antibody-producing cells are key therapeutic targets. Antibodies are produced by antigen-specific B-cell subsets, that is, short-lived plasmablasts and long-lived plasma cells. The B-cells also require T-cell help for activation, and thus MG is also classified as a B-cell mediated, T-cell dependent autoimmune disease. Nevertheless, our understanding of the cellular immunology in MG is still incomplete. Every B-cell undergoes V(D)J recombination that stochastically generates one type of B-cell receptor (BCR)/antibody, resulting in up to 10^11^ unique B-cell clones, of which some will recognize self-antigens and may cause autoimmunity. Therefore, during maturation, B-cells undergo negative selection for self-antigens in the bone marrow at two “tolerance checkpoints” to remove poly- and autoreactive B-cells by clonal deletion, induction of anergy, or receptor editing [223,224,225]. The BCR repertoire in MG was found to be disturbed in a small number of patients [226], suggesting that aberrant B-cell receptor editing might contribute to the formation of autoreactive B-cells. Furthermore, it was shown that in MG patients, defects in the negative selection lead to an escape of autoreactive B-cells from checkpoint controls [227]. In line with this observation, patients with MG have a high risk in developing further autoimmune diseases (poly-autoimmunity) [228]. At the same time, the patients also have reduced numbers of B10 cells, which is a subset of regulatory B-cells (Bregs) [229,230].

Nevertheless, B-cells depend on activation by T-cells that recognize the same antigen, and the negative selection of autoreactive T-cells is considered as a more stringent process. T-cells undergo a negative selection for self-antigens in the thymus, specifically in the thymic medulla. Here, the autoimmune regulator (AIRE) transcription factor, expression of which itself is regulated by estrogen [231,232], induces the ectopic expression of tissue-restricted self-antigens in stromal cells such as medullary thymic epithelial cells (mTECs). These antigens are then presented by mTECs directly or after antigen “handover” by antigen-presenting cells to the developing T-cells [233,234]. Upon a strong binding of T-cells via the T-cell receptor to self-antigens, the cells are removed from the repertoire, by clonal deletion, induction of anergy, or by a transformation to regulatory T-cells (Tregs) that have an immune regulatory and anti-inflammatory function. It is thought that thymic abnormalities such as follicular hyperplasia or thymoma contribute to defects in central tolerance mechanisms and the escape of AChR specific T-cells [235]. These include: (1) a defective AIRE expression in mTECs [236,237]; (2) the downregulation of AIRE transcription by estrogen, which may explain the female predominance [232,238]; (3) an absence of thymic medulla and cells involved in the negative selection in thymoma [239,240,241,242]; (4) hyperplastic or neoplastic mTECs presenting autoantigen [243,244] or downregulating MHC II expression [239,245,246], and (5) aberrant expression of pro-inflammatory cytokines (reviewed by [235,247]).

These mechanisms may contribute to the escape of AChR-specific T helper (Th) cells (CD4+ cells) from tolerance that then activate AChR-specific B-cells to produce autoantibodies. Thymic myoid cells express muscle antigens, among these is the whole intact AChR. Antibody and complement attack on thymic myoid cells may lead to the formation of antigen-antibody complexes that stimulate antigen-presenting cells and lead to the formation of ectopic germinal centers (GC) in the thymus [248,249,250,251,252]. Ectopic GCs in the EOMG thymus promote antigen-driven affinity maturation and differentiation to autoreactive memory B-cells and plasma cells [253]. B-cells in the thymus of MG patients with hyperplasia showed an increase in memory B-cells and potentially autoreactive B-cells associated with autoimmunity [242,254]. B-cells in the MG thymus were found to be activated [255], clonally expanded, and also phenotypically more different compared to B-cells in healthy control thymi [256,257]. Ectopic GCs in the thymus contained AChR-specific long-lived plasma cells [258,259,260] that spontaneously produced high levels of AChR antibodies in vitro [260,261,262,263,264,265,266]. Plasmablast levels were also elevated in the blood of patients with AChR MG, though it was not tested whether these produced myasthenic antibodies [242]. Thymic cells derived from patients with Lrp4 MG also produced high levels of IgG, but Lrp4-specific antibodies could not be detected [50].

AChR-specific Th cells are important in MG to induce the production of the pathogenic AChR antibodies [267,268]. The altered thymic function causes an increase in the release of autoreactive CD4+ and CD8+ T-cells and leads to altered T-cell subset balance [239,241,268,269,270,271]. T follicular helper cells, which play a role in B-cell selection and survival in GCs, were increased in numbers [272,273]. Th17 cells play a role in inflammation and tissue injury in autoimmunity [274], and may also play a role in MG [275,276,277,278,279,280]. Furthermore, an increase in the Th1/Th17 cells was observed in MG patients, as well as a change in the balance of cytokines, with increased levels of Th1/Th17 cytokines and reduced levels of IL-10 [229,278,281,282,283]. Tregs are key players in the maintenance of peripheral self-tolerance [284,285,286]. In MG, Tregs were functionally impaired and unable to suppress T-cell activation in vitro [277,287,288,289,290] or they were found to be reduced in numbers [271,291,292,293].

In addition to the abnormalities in the immune cells, other factors may contribute to the early stages of the pathogenesis, including virus infection, genetic predisposition, environmental factors (e.g., stress, vitamin D levels), dysregulation of pro-inflammatory cytokines, chemokines and miRNAs, aberrant expression of Toll-like receptors, and others, but they exceed the limits of this review. The following excellent reviews are suggested for further reading: [48,235,247,294,295,296,297].

## 6. Therapies Targeting the NMJ

### 6.1. Therapies Targeting the Immune System

MG is an autoimmune disease caused by autoantibodies and as such, treatment relies on targeting the immune system to reduce pathogenic antibodies [298,299,300,301]. This includes general immunosuppression, removal of antibodies by plasmapheresis [302], modulating the immune system by intravenous IgG [303], and the removal of the thymus, where antibody producing cells reside [40].

More recent approaches include the use of complement inhibitors, specifically monoclonal antibodies targeting C5 (eculizumab), have been widely studied as a potential treatment and are considered as clinically beneficial for MG patients [304,305,306,307], and there are efforts to develop new monoclonal antibodies or small interfering RNAs (siRNAs) targeting the complement system [195,308,309,310]. A different approach is the direct linking of a complement regulator to the motor endplate by fusion to single chain AChR antibodies [196]. Another line of treatment focuses on the depletion of antigen producing cells. Current models propose that in AChR MG antibody production relies mostly on long-lived plasma cells. Removal of the thymus, which is the site where long-lived plasma cells reside, is clinically beneficial for the patients [40]. Another strategy to remove plasma cells is the use of proteasome inhibitors like Bortezomib, which was proven to be efficient in vitro and in vivo [311,312].

In contrast, B-cell depletion therapy with monoclonal antibodies against CD20 (Rituximab) is less efficient for the removal of long-lived plasma cells because CD20 is only expressed on B-cells and precursors for short-lived plasma cells [313], but not long-lived plasma cells [314,315]. Patients with MuSK MG benefit from B-cell depletion more than patients with AChR MG, which suggests that MuSK antibodies are produced by short-lived plasmablasts [316,317,318,319,320,321,322,323,324,325,326,327,328,329]. In addition to Rituximab, a range of new monoclonal antibodies targeting B-cells, antibodies and cytokines are being developed [300,330].

Taken together these therapies do, however, not target antigen-specific antibodies or antibody producing cells, but target whole parts of the immune system. New, bold approaches are necessary for tailored, antigen-specific therapy. This includes immunization with cytoplasmic parts of the AChR (similar to a vaccination), to induce the production of antibodies that are non-pathogenic [331], which was found to be beneficial in a MG animal model. Apheresis of antigen-specific antibodies or antigen-specific plasma cells are further tailored therapies [332,333], as is a competitive block of the binding site for the AChR antibody with a monoclonal antibody (mAB). The mAb 637 was cloned from an AChR MG patient, and with IgG1 subclass-induced MG-like symptoms in rhesus monkeys, but its immunologically inert and hinge-deleted IgG4 isoform competed with pathogenic antibodies and ameliorated the symptoms in the animals [334].

### 6.2. Therapies Targeting the NMJ

A different approach is to target the structure and function of the NMJ. Most patients are treated routinely with AChE inhibitors (AChEi) [335], which prolong the effector time of ACh at the synapse and compensate for reduced AChR densities at the synapse. Notably this is not the case in MuSK MG, where patients experience exacerbation with AChEi [336,337], which suggests an increased sensitivity to ACh in MuSK MG. Presynaptic changes can be found in MuSK MG patients and, unlike in AChR MG patients, their quantal release of ACh is not increased [31,32], perhaps due to a block of retrograde signaling by the MuSK antibodies. It might be possible that other factors that are released by the motor neuron, such as Agrin, might also be reduced, depleting positive signals for NMJ maintenance. The prolonged dwell of ACh in the synaptic cleft could in theory then destabilize AChR clusters, as ACh is also a signal for AChR cluster dissipation [338,339]. There are also observations that MuSK MG patients respond with repeated action potentials to AChEi, probably due to a prolonged EPP and lack of Na channel inhibition [24,340]. Furthermore, MuSK also has a role for anchoring ColQ to the synapse and one group found an effect of MuSK antibodies on ColQ-MuSK interaction [341], suggesting reduced levels of AChE at the MuSK MG synapse.

For patients with MuSK MG, strengthening the NMJ architecture, and specifically the dense clustering of AChR, could be a new treatment strategy. MuSK antibodies block phosphorylation of MuSK, which in turn leads to reduced MuSK activation and signal transduction. Experiments by Huda et al. showed that the inhibition of the tyrosine phosphatase Shp2 overcame the effects of MuSK antibodies on AChR clusters in vitro [342]. In line with this observation is that a loss of Dok-7, which is also important for MuSK activation, might increase the susceptibility to MG in experimental animals [343]. Casein kinase 2 (CK2) was also found to induce serine phosphorylation in the MuSK kinase domain, which promotes AChR clustering [344] and may also be a target for therapies. Another attempt to fortify the NMJ against the effects of MG antibodies is the overexpression of rapsyn, which stabilized the AChRs and successfully prevented the loss of AChRs in animal models [345,346]. In a different approach, a recombinant Agrin consisting of the C-terminal region of mouse neural Agrin was found to improve MG symptoms and neurotransmission and to promote AChR clustering in a rat active immunization model of MG [347]. These studies, though still in early experimental phases, show great promise, and it would be important to improve research into these NMJ-specific therapies. To this end, it would be important to develop a physiological model of the human NMJ such as in vitro NMJs generated from human myoblasts and motor neurons. Efforts to develop such a model system are being made using human-induced pluripotent stem cells (iPSCs). Although these studies have demonstrated proper differentiation into motor neurons and myotubes, as well as nerve-dependent contraction and calcium influx, characteristic features of pre-and postsynaptic development are still lacking [348,349,350,351]. Therefore, reliable in vitro models recapitulating functional human NMJs are still unavailable.

## 7. Challenges of Antibody Diagnostics

There are several commercially available reliable diagnostic tests to detect autoantibodies against AChR and MuSK (these include radioimmunoassays and ELISA). In a subgroup of patients, however, autoantibodies against AChR cannot be detected by these assays. These antibodies recognize clustered AChR and can only be detected by cell-based assays (CBAs) [38,352,353,354]. Overall, CBAs are perhaps the most sensitive method of choice for the detection of antibodies against cell-surface expressed structures, if appropriate controls (mock transfected cells, control sera) are included in the assay [38,355,356,357,358]. These are semi-quantitative by fluorescence microscopy, but also quantitative if binding of antibodies to the antigen-transfected cells is measured by flow-cytometry [211]. Diagnostic methods for the detection of autoantibodies against AChR, MuSK, and clustered AChR are well established. This is unfortunately not yet the case for all myasthenic antibodies. The frequency of Lrp4 MG varies between 2–50% in the group of patients without AChR and MuSK antibodies, depending on the method of choice for detecting autoantibodies [38,48,75,359]. One difficulty is the expression of Lrp4 and specifically the transport to the cell surface for CBAs, which might be slightly enhanced by co-expression of chaperones such as Mesdc2 [360], or by using fixed and permeabilized cells. The former approach might still lead to low levels of antigen expression and an underestimation of antibody levels, while the latter approach cannot exclude the binding of antibodies to irrelevant intracellular antigen, such as Lrp4s (and other proteins) still in the endoplasmic reticulum, that have not been properly folded and/or glycosylated. The variation of frequencies with live and fixed CBAs as well as ELISA demonstrates that specificity and sensitivity of these assays need optimization and cross-validation over several centers.

Lrp4 and Agrin antibodies have also been detected in patients with amyotrophic lateral sclerosis (ALS) [361,362,363], and their pathogenic role in MG and ALS is unclear. Notably, AChR antibodies were also detected in ALS patients [363,364,365]. AChR antibody pathogenicity in the context of MG is well documented, but it remains unclear in ALS, since passive transfer models using serum or IgG from AChR, as well as Lrp4 and Agrin antibody-positive ALS patients, have not yet been reported. It is possible that these antibodies are epiphenomena or they play a role in both diseases, and perhaps they have diagnostic value as biomarkers.

In any case, we need improved methods to reliably detect the full range of myasthenic autoantibodies.

## 8. Concluding Remarks

Our understanding of MG has greatly improved in the last few years and decades, especially regarding the pathogenic mechanisms of AChR and MuSK IgG4 antibodies, but it is far from complete. The etiology and immunopathogenesis are unclear, as is the contribution of MuSK IgG1–3 and antibodies against Lrp4, Agrin, and ColQ for disease. Most importantly, a fraction of 5–10% of patients have no known autoantibodies and are still considered as seronegative. The discovery of new autoantigens at the NMJ in these patients is essential for the understanding of their pathophysiology. New and improved diagnostic assays for Lrp4, Agrin, and any new autoantigens that may be identified are necessary for correct diagnosis and clinical management. Most importantly, we will require an improved in vitro model of the NMJ to recreate the physiological environment in MG, for antigen discovery studies, but also to study the pre- and postsynaptic pathogenic mechanisms of MG antibodies and for the development of new therapeutic strategies at the patient NMJ.

## Figures and Tables

**Figure 1 cells-08-00671-f001:**
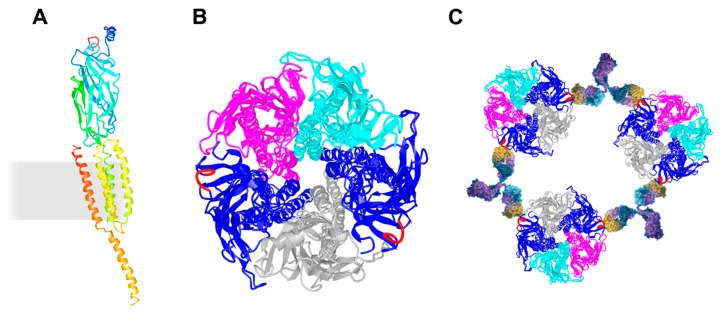
Schematic presentation of the AChR and its cross-linking via the MIR. (**A**) Ribbon diagram of a single AChR α subunit viewed in an orientation such that the central axis of the pentamer is to the side (right). The membrane is indicated as a grey rectangle. α-helices M1–M4 are membrane-spanning and form the ion-permeable gate of the AChR. The MIR (shown in red) is located between β2- and β3-strands. (**B**) Ribbon diagram of the whole receptor viewed from the synaptic cleft. AChR α in blue with the MIR in red, AChR β in magenta, AChR δ in cyan, and AChR γ in grey. The MIR is shown in red. (**C**) Schematic presentation of autoantibody-induced inter-molecular cross-linking via the MIR. The molecular models were drawn using the molecular coordinates from the density maps deposited in the Protein Data Bank (PDB file accession number 3BG9 [112]) and the Web-based 3D Structure Viewer iCn3D.

**Figure 2 cells-08-00671-f002:**
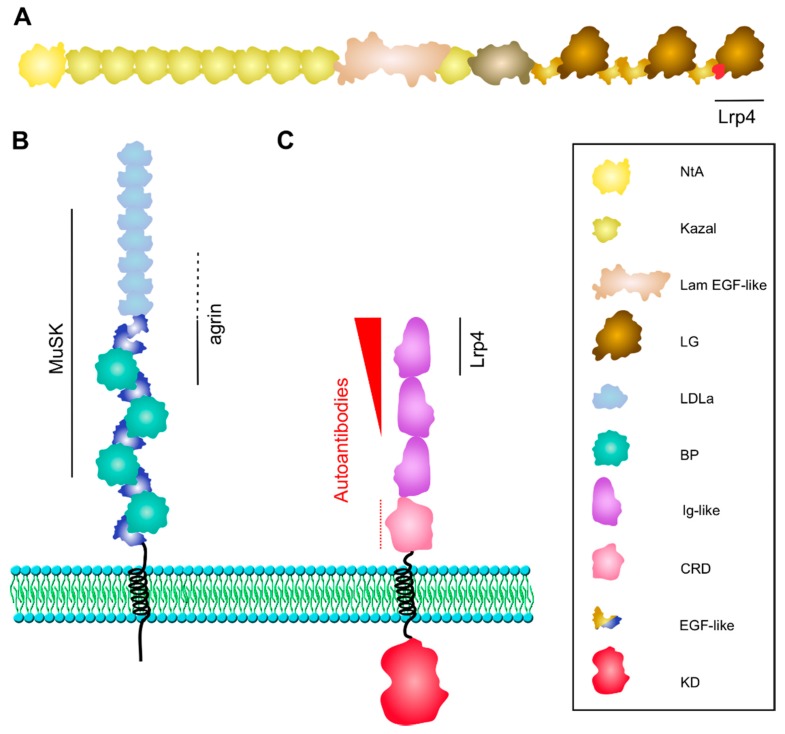
Structure of Agrin, Lrp4, and MuSK. (**A**) Neuronal Agrin is anchored in the basal lamina via its laminin-binding NtA-domain. The third LG domain contains the B/Z splice insert (in red), which forms a loop critical for Lrp4 binding. (**B**) Lrp4 is comprised of a large extracellular domain, one transmembrane domain and a short cytoplasmic tail. The BP domains 1–3 and LDLa repeats 4–8 are sufficient for binding to MuSK. The first BP domain is critical for Agrin binding with supporting function of LDLa repeats 6–8 (dotted line). (**C**) MuSK interacts with Lrp4 via the Ig1 domain. Ig1 is also important for homodimerization. The kinase domain is phosphorylated and activated upon Agrin-induced binding of Lrp4. MuSK autoantibodies are predominantly directed against the Ig1 domain, less against the Ig2 domain (shown as red arrowhead), and occasionally against the CRD (dotted line). BP, β-propeller; CRD, cysteine-rich domain; LDLa, low-density lipoprotein receptor domain class A; KD, kinase domain; Lam EGF-like, laminin EGF-like; LG, laminin globular-like; NtA, N-terminal Agrin.

**Figure 3 cells-08-00671-f003:**
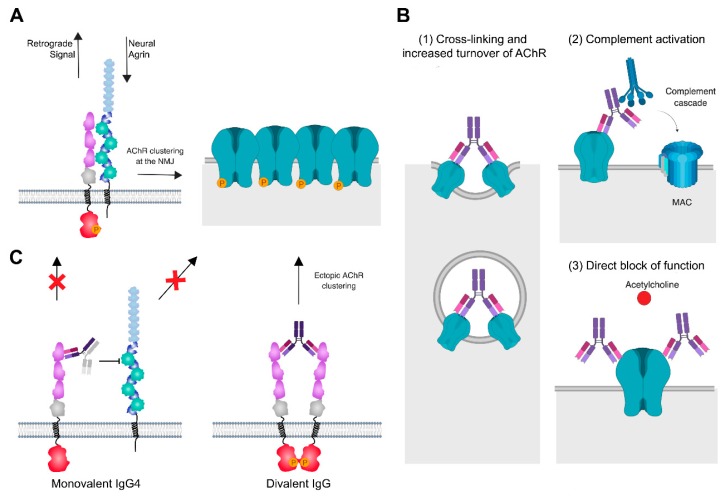
Pathogenic mechanisms of MG autoantibodies at the NMJ. (**A**) At the healthy NMJ, neural Agrin stimulation induces interaction between Lrp4 and MuSK, leading to MuSK autophosphorylation and activation and the phosphorylation and clustering of AChRs. A retrograde signal for presynaptic development is sent via Lrp4. (**B**) MG antibodies of IgG1 and IgG3 subclass against AChR have three pathogenic mechanisms: (1) Cross-linking and increased turnover of AChR lead to reduced AChR levels at the NMJ, (2) activation of the classical complement cascade, formation of the membrane attack complex (MAC) and complement-mediated damage of the postsynaptic membrane, and (3) direct block of function by preventing the binding of acetylcholine. (**C**) Bispecific IgG4 antibodies of IgG4 subclass against MuSK bind monovalently to MuSK and block Lrp4-MuSK interaction, thus interrupting the agrin-Lrp4-MuSK-Dok7 signaling axis and causing reduced densities of AChR at the synapse. A further effect is the disruption of a retrograde signal from Lrp4 to the motor neuron. Divalent binding of MuSK IgG leads to dimerization, autophosphorylation, and activation of MuSK independent of Agrin stimulation and causes the formation of ectopic AChR clusters. Created with BioRender.

**Table 1 cells-08-00671-t001:** Clinical subgroups of myasthenia gravis (modified from [4]).

MG Subgroup	Clinical Characteristics	Antigen	Thymus Pathology	IgG Subclass
Early onset MG (EOMG)	Age of onset <50, sex ratio (F:M) 3:1, genetic association with HLA-B8, A1, and DRw3	AChR	Thymic lymphofollicular hyperplasia	IgG1, IgG
Late onset MG (LOMG)	Age of onset >50, sex ratio (F:M) 1:1.5, genetic association with HLA-A3, B7, and DRw2	AChR	Normal thymus (age-related thymus atrophy)	IgG1, IgG3
Thymoma associated MG (TAMG)	Paraneoplastic MG, non-pathogenic antibodies against striated muscle, titin, ryanodine receptor	AChR	Thymoma	IgG1, IgG3
Ocular MG (OMG)	Restricted to ocular muscles, low AChR titres	AChR	Variable, no lymphoid follicles	IgG1, IgG3
MuSK MG	Severe phenotype, respiratory, and bulbar muscle weakness, sex ratio (F:M) up to 9:1, genetic association with HLA-DR14-DQ5	MuSK	Normal thymus	IgG4
Lrp4 MG	Mild phenotype, sex ratio (F:M) 2.5:1	Lrp4	Variable (normal, thymoma, thymic lymphofollicular hyperplasia)	IgG1, IgG2
Agrin MG	Generalized weakness, often also additional AChR, MuSK, or Lrp4 antibodies, associated with severe weakness	Agrin	No thymoma (few studies)	N/A
Transient neonatal MG (TNMG)	Mild symptoms, onset at birth, remission after days to months	AChR, MuSK		maternal IgG
Fetal myasthenia gravis	Reduced fetal mobility, arthrogryposis congenital (AMC), very severe, risk of fetal death	Fetal AChR γ subunit		maternal IgG

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
