# Peer review of "Myasthenia Gravis: Pathogenic Effects of Autoantibodies on Neuromuscular Architecture"

_cells, 2019, doi:10.3390/cells8070671_

Round 1
Reviewer 1 Report
This is an excellent, well-written review on the antibody-mediated pathogenic mechanisms of myasthenia gravis. While there have been a number of reviews on this topic this one stands out by providing the reader with a deeper understanding of the antigens and how the antibodies target their function. However, there are a number of issues that need to be addressed before the manuscript is ready for publication, as listed below.
Lines 114-123 might give the non-expert (i.e. the intended reader) the impression that the pathogenic nature of anti-LRP4 and anti-agrin is well established, or that it will be very soon. Maybe, but in the autoimmunity field there have been a lot of misleading, bystander antibodies and it is important to rule out this (less interesting) hypothesis with each new autoantibody. The first section indicates that the AChR and MuSK seropositive forms of MG fulfill the Witebsky postulates but it doesn’t outline what these postulates are nor does the manuscript properly spell out properly what evidence is still required to confirm that anti-LRP4 and anti-agrin seropositive cases really are caused by these antibodies. Not everything that seems intuitively obvious turns out to be true: in vitro experiments and active immunization studies in animals are not sufficient proof. This section should be modified to make the uncertainty explicit.
It might be useful to the reader to include an outline of the typical evidence basis for a diagnosis of MG which, as I understand it, involves weighing clinical and electrophysiology as well as seropositivity.
Line 290. Since the isoform of agrin found throughout the skeletal muscle basement membrane does not activate LRP4-MuSK, remind the reader that you are referring to the neural splicing form of agrin when describing NMJ development.
Line 322 “In addition, phosphates [tyrosine phosphatases?] such as Shp2”
Box 2 describes the comparative structure/function of various IgG isoforms and proteolytic fragments. For the non-immunologist this box would be improved by adding a cartoon figure illustrating the structures of the IgGs.
Line 373-382 this is the first time (evoked) EPP is mentioned. Some brief description of quantal neuromuscular transmission is needed to explain the EPP and quantal content, in relation to neurotransmission and the MEPP. Also explain what ‘safety factor’ means, its significance and how impaired safety factor can be detected by CMAP decrement. These concepts are not intuitive and need to be explained, albeit briefly, to anyone new to this field. Concerning the feedback regulation of quantal content, there is some, more recent, work by Mark Rich’s group (Wang et al., 2016) and a recent review article (Davis & Müller, 2015).
Davis GW & Müller M. (2015). Homeostatic control of presynaptic neurotransmitter release. Annu Rev Physiol 77, 251-270. .
Wang X, Pinter MJ & Rich MM. (2016). Reversible Recruitment of a Homeostatic Reserve Pool of Synaptic Vesicles Underlies Rapid Homeostatic Plasticity of Quantal Content. J Neurosci 36, 828–836.
Line 576 “Prolonged ACh incubation [perhaps instead: ‘dwell in the synaptic cleft’]”
Section 5.3 seems to take for granted that the anti-LRP4, anti-agrin and anti-ColQ antibodies cause myasthenia in patients that possess these antibodies. This section needs to spell out the nature of the current experimental evidence that anti-LRP4 etc. can interfere with MuSK signaling and cause myasthenia. The reader is not informed of this uncertainty until this it is mentioned in passing near the end of the review (sections 7 & 8). What kind of experiments with what kind of anti-LRP4 antibodies (addition of serum to cultured mouse myotubes blocks agrin-induced ACHR clustering? Inhibition of agrin-induced MuSK kinase activation? Active immunization of rabbits with LRP4?...). This is the most controversial part of the review but also the weakest part.
6. Therapies targeting the NMJ 6.1. Current strategies: this section is oddly titled. It focuses instead only on immunosuppression and targeting of B cells and probably should mention cholinesterase inhibitors given the title “Therapies targeting the NMJ”. It also might consider mentioning pilot studies targeting complement inhibitors to motor endplates ( Kusner LL, Satija N, Cheng G & Kaminski HJ. (2014). Targeting therapy to the neuromuscular junction: proof of concept. Muscle Nerve 49, 749-756.)
Author Response
REVIEWER1
Comments and Suggestions for Authors
We thank the reviewer for the in-depth study of our manuscript and for the constructive comments, which clearly helped to improve the quality of the manuscript. For the revision we added additional information regarding electrophysiology, provide a new illustration for Box2 and try to clarify the current status of research on anti-Lrp4/anti-agrin antibodies. All concerns are addressed point-by point below.
(1) Lines 114-123 might give the non-expert (i.e. the intended reader) the impression that the pathogenic nature of anti-LRP4 and anti-agrin is well established, or that it will be very soon. Maybe, but in the autoimmunity field there have been a lot of misleading, bystander antibodies and it is important to rule out this (less interesting) hypothesis with each new autoantibody.
To address this remark, we have reworded the section (line 183-189) and also referenced Chapter 5.3 in which the antibodies are discussed in detail.
(2) The first section indicates that the AChR and MuSK seropositive forms of MG fulfill the Witebsky postulates but it doesn’t outline what these postulates are nor does the manuscript properly spell out properly what evidence is still required to confirm that anti-LRP4 and anti-agrin seropositive cases really are caused by these antibodies. Not everything that seems intuitively obvious turns out to be true: in vitro experiments and active immunization studies in animals are not sufficient proof. This section should be modified to make the uncertainty explicit.
As suggested we have now outlined the criteria for the Witebsky postulate. In addition, we have included a sentence describing why anti-Lrp4 and anti-agrin antibodies do not fulfill these criteria. Line 94-104
(3) It might be useful to the reader to include an outline of the typical evidence basis for a diagnosis of MG which, as I understand it, involves weighing clinical and electrophysiology as well as seropositivity.
We have included a brief outline on the diagnosis of MG. However, we are molecular biologists and not physicians, therefore we deem it more appropriate to keep the information on the clinical diagnostic strategy concise, and rather refer the reader to several recent clinical reviews on MG. We do discuss challenges of the autoantibody diagnostics in detail towards the end of the manuscript, since these are limited to the laboratory tests that are part of our experimental repertoire. Line 49-58
(4) Line 290. Since the isoform of agrin found throughout the skeletal muscle basement membrane does not activate LRP4-MuSK, remind the reader that you are referring to the neural splicing form of agrin when describing NMJ development.
We have now specifically stated that the neuronal isoform of agrin is required for MuSK activation (line 356).
(5) Line 322 “In addition, phosphates [tyrosine phosphatases?] such as Shp2”
As suggested we changed phosphatase to the more specific term tyrosine phosphatase (line 386).
(6) Box 2 describes the comparative structure/function of various IgG isoforms and proteolytic fragments. For the non-immunologist this box would be improved by adding a cartoon figure illustrating the structures of the IgGs.
We have now included a cartoon visualizing the most important feature of IgG molecules.
(7) Line 373-382 this is the first time (evoked) EPP is mentioned. Some brief description of quantal neuromuscular transmission is needed to explain the EPP and quantal content, in relation to neurotransmission and the MEPP. Also explain what ‘safety factor’ means, its significance and how impaired safety factor can be detected by CMAP decrement. These concepts are not intuitive and need to be explained, albeit briefly, to anyone new to this field. Concerning the feedback regulation of quantal content, there is some, more recent, work by Mark Rich’s group (Wang et al., 2016) and a recent review article (Davis & Müller, 2015).
Davis GW & Müller M. (2015). Homeostatic control of presynaptic neurotransmitter release. Annu Rev Physiol 77, 251-270. .
Wang X, Pinter MJ & Rich MM. (2016). Reversible Recruitment of a Homeostatic Reserve Pool of Synaptic Vesicles Underlies Rapid Homeostatic Plasticity of Quantal Content. J Neurosci 36, 828–836.
The revised version of our manuscript now contains a section in Chapter 1 describing safety factor, quantal content and how these affect CMAP in MG. Line 59-93
In addition, we have included the suggested references.
(8) Line 576 “Prolonged ACh incubation [perhaps instead: ‘dwell in the synaptic cleft’]”
The sentence was changed to “Prolonged dwell of ACh in the synaptic cleft could…” Line 678
(9) Section 5.3 seems to take for granted that the anti-LRP4, anti-agrin and anti-ColQ antibodies cause myasthenia in patients that possess these antibodies. This section needs to spell out the nature of the current experimental evidence that anti-LRP4 etc. can interfere with MuSK signaling and cause myasthenia. The reader is not informed of this uncertainty until this it is mentioned in passing near the end of the review (sections 7 & 8). What kind of experiments with what kind of anti-LRP4 antibodies (addition of serum to cultured mouse myotubes blocks agrin-induced ACHR clustering? Inhibition of agrin-induced MuSK kinase activation? Active immunization of rabbits with LRP4?...). This is the most controversial part of the review but also the weakest part.
We have improved this section to clarify the current knowledge on Lrp4, Agrin and ColQ antibodies (Chapter 5.3)
(10) 6. Therapies targeting the NMJ 6.1. Current strategies: this section is oddly titled. It focuses instead only on immunosuppression and targeting of B cells and probably should mention cholinesterase inhibitors given the title “Therapies targeting the NMJ”. It also might consider mentioning pilot studies targeting complement inhibitors to motor endplates ( Kusner LL, Satija N, Cheng G & Kaminski HJ. (2014). Targeting therapy to the neuromuscular junction: proof of concept. Muscle Nerve 49, 749-756.)
We have addressed this remark by changing the sections to discuss: 6.1 Therapies targeting the immune system and 6.2 Therapies targeting the NMJ. We also have added the appropriate references. Cholinesterase inhibitors are also discussed.

Reviewer 2 Report
Myasthenia gravis: pathogenic effects of autoantibodies on neuromuscular architecture
The review provides an extensive summary of the autoantibodies that cause MG.
The review focuses on the subtypes of MG, the history, the antigens at the NMJ, the pathogenic effects of the autoantibodies, the immune production of the autoreactive cells and production of the autoantibodies, the therapeutic strategies, and the diagnostic testing for determining the autoantibody expression.
The below detail the specific on suggested changes to the manuscript:
Line 40: “AChR MG patients, ocular MG (OMG) with predominantly ocular symptoms and (4) fetal or neonatal” requires the placement of (4) before ocular MG and change the (4) to a (5) for the fetal or neonatal>
Line 175: The paragraph beginning “The N-terminal region of AChRα represents the main immunogenic region (MIR)” requires citations.
Line 359: “Here, C1q binds to the Fc region of the AChR antibodies, followed by downstream products of the complement cascade such as C3” – The sentence needs to be reworded. C3 is not a downstream product but the breakdown of C3 into C3a and C3b is a product of complement.
Line 363: “Furthermore, inhibition of complement cascade activation by suppressing the expression of C2, significantly improved clinical symptoms in a mouse model of MG [164].” Several studies have been done to confirm the complement attack to the neuromuscular junction is significant. The authors should generalize the statement and include the references for other studies.
Line 370: “Direct block of function of the AChRs by preventing binding of acetylcholine [49,169-172].” Also include the antibody binding can block the channel.
Line 374: “The reduced levels of voltage gated sodium channels in turn lead to an increased threshold to induce an action potential” The reference is incorrect. 173. Phillips, W.D.; Vincent, A. Pathogenesis of myasthenia gravis: update on disease types, models, and mechanisms. F1000Research 2016, 5, doi:10.12688/f1000research.8206.1 should be changed to Ruff RL, and Lennon VA, How Myasthenia Gravis Alters the Safety Factor for Neuromuscular Transmission. J Neuroimmunol. 2008 Sep 15; 201-202: 13–20.
Line 384: “The γ subunit is only expressed during the first 30 weeks of life” Not all AChR change to the epsilon subunit. The neuromuscular junctions of the extraocular muscles maintain the gamma subunit.
Section on NMJ aging can be integrated into the review with the suggestion that the fragmentation may be aiding the LOMG. A section that could be added is EMG recordings.
Author Response
REVIEWER2
Comments and Suggestions for Authors
We thank the reviewer for the helpful and constructive criticism. The revised manuscript includes additional information on EMG and the role of complement in MG as well as adjusted the references. We have addressed the raised points and provide a detailed reply below.
(1) Line 40: “AChR MG patients, ocular MG (OMG) with predominantly ocular symptoms and (4) fetal or neonatal” requires the placement of (4) before ocular MG and change the (4) to a (5) for the fetal or neonatal
As suggested the order was changed accordingly. Line 41
(2) Line 175: The paragraph beginning “The N-terminal region of AChRα represents the main immunogenic region (MIR)” requires citations.
We have now included references for the MIR: its first description (PMID: 6153804), the localization by EM (PMID: 3611197) and mutagenesis (PMID: 3611197). Line238
(3) Line 359: “Here, C1q binds to the Fc region of the AChR antibodies, followed by downstream products of the complement cascade such as C3” – The sentence needs to be reworded. C3 is not a downstream product but the breakdown of C3 into C3a and C3b is a product of complement.
We have now rephrased the sentence to “Here, C1q binds to the Fc region of the AChR antibodies, followed by the breakdown of C3 into C3a and C3b, which as downstream products of the complement cascade initiate the formation of the membrane attack complex.” Further, we provide additional information on the role of complement in MG. Line 425-435
(4) Line 363: “Furthermore, inhibition of complement cascade activation by suppressing the expression of C2, significantly improved clinical symptoms in a mouse model of MG [164].” Several studies have been done to confirm the complement attack to the neuromuscular junction is significant. The authors should generalize the statement and include the references for other studies.
We have added more evidence for the pathogenic role of complement to Chapter 5.1 and also added more information about Eculizumab in Chapter 6.1. (line 435-440 and line 641-623).
(5) Line 370: “Direct block of function of the AChRs by preventing binding of acetylcholine [49,169-172].” Also include the antibody binding can block the channel.
We have now rephrased the sentence to “Direct inhibition of the function of the AChRs by preventing binding of acetylcholine or blocking of the channel [49,169-172].” Line 445-456
(6) Line 374: “The reduced levels of voltage gated sodium channels in turn lead to an increased threshold to induce an action potential” The reference is incorrect. 173. Phillips, W.D.; Vincent, A. Pathogenesis of myasthenia gravis: update on disease types, models, and mechanisms. F1000Research 2016, 5, doi:10.12688/f1000research.8206.1 should be changed to Ruff RL, and Lennon VA, How Myasthenia Gravis Alters the Safety Factor for Neuromuscular Transmission. J Neuroimmunol. 2008 Sep 15; 201-202: 13–20.
We have now adjusted this section. CMAP in AChR MG is described in detail in Chapter 1 (line 79-87). Here we have also added the suggested reference.
(7) Line 384: “The γ subunit is only expressed during the first 30 weeks of life” Not all AChR change to the epsilon subunit. The neuromuscular junctions of the extraocular muscles maintain the gamma subunit.
We have now referred to the observation that AChR gamma expression is maintained (PMID: 7684117). Line 455-456
(8) Section on NMJ aging can be integrated into the review with the suggestion that the fragmentation may be aiding the LOMG. A section that could be added is EMG recordings.
The section on NMJ aging (4.4) was complemented with a note about the possibility, that aging contributes to LOMG. Line 412-414
As suggested we have now added a section in Chapter 1 describing electrophysiology of the NMJ including quantal content, safety factor and CMAP. Line 59-93
Round 2
Reviewer 1 Report
The authors have addressed my concerns